# BLIND BASELINES BEAT MEMBERSHIP INFERENCE ATTACKS FOR FOUNDATION MODELS

## ABSTRACT

Membership inference (MI) attacks try to determine if a data sample was used to train a machine learning model. For foundation models trained on unknown Web data, MI attacks are often used to detect copyrighted training materials, measure test set contamination, or audit machine unlearning. Unfortunately, we find that evaluations of MI attacks for foundation models are flawed, because they sample members and non-members from different distributions. For 9 published MI evaluation datasets, we show that *blind* attacks—that distinguish the member and non-member distributions without looking at any trained model—outperform state-of-the-art MI attacks. Existing evaluations thus tell us nothing about membership leakage of a foundation model's training data.

## 1 INTRODUCTION

Many foundation models (Bommasani et al., 2021) such as GPT-4 (OpenAI, 2023), Gemini (Gemini Team, 2023) and DALL-E (Ramesh et al., 2021) are trained on unknown data. There is great interest in methods that can determine if a piece of data was used to train these models. Such methods—called *membership inference attacks* (Shokri et al., 2017)—have been studied for foundation models in many recent works (Shi et al., 2023; Duan et al., 2024; Ko et al., 2023; Dubiński et al., 2024; Zhang et al., 2024a; Meeus et al., 2023). Applications include privacy attacks (Carlini et al., 2021), demonstrating the use of copyrighted material (Meeus et al., 2024b), detecting test data contamination (Oren et al., 2023), or auditing the efficacy of methods to "unlearn" training data (Bourtoule et al., 2021).

To evaluate the performance of a membership inference attack, it is common to train a model on a random subset of a larger dataset, and then ask the attacker to distinguish these *members* from the remaining *non-members* by interacting with the model. Emulating this setup for foundation models is hard, since we often do not have access to a held out set sampled from the *same distribution* as the training set.

Existing MI evaluations thus create member and non-member datasets *a posteriori*, typically by picking members from sources known or suspected to be in the targeted foundation model's training set, and then attempting to emulate the distribution of these sources to sample non-members.

Unfortunately, we show these strategies are severely flawed, and create easily distinguishable member and non-member distributions. As a special case of this flaw, concurrent work of Duan et al. (2024), Maini et al. (2024) and Meeus et al. (2024a) finds a temporal shift between members and non-members in the Wiki-MIA dataset (Shi et al., 2023). We show this issue is persistent by identifying significant distribution shifts (beyond temporal shifts) in 9 MI evaluation datasets for foundation models, for both text and vision. Worse, we show that existing MI attacks perform "worse than chance" on these datasets. Specifically, we design "blind" attacks, *which completely ignore the target model*, and outperform all reported results from state-of-the-art MI attacks (see Table 1).

Our methods are naive: for some datasets with a temporal shift, we apply a threshold to specific dates extracted from each sample; for other text or text-vision datasets, we build simple bag-of-words or n-gram classifiers on captions. We find that such simple methods work even for MI evaluation datasets that were explicitly designed to remove distribution shifts between members and non-members. Our work shows that removing biases in a post-hoc fashion is highly brittle.

Table 1: **MI evaluation datasets for foundation models are flawed.** Our *blind* attacks distinguish members from non-members better than existing attacks, without looking at any model.

| MI dataset | Metric | Best Reported (%) | Ours (%) |
|---|---|---|---|
| WikiMIA | TPR@5%FPR | 43.2 | **94.7** |
| BookMIA | AUC ROC | 88.0 | **91.4** |
| Temporal Wiki | AUC ROC | 79.6 | **79.9** |
| Temporal ArXiv | AUC ROC | 74.5 | **75.6** |
| ArXiv (all vs 1 month) | TPR@1%FPR | 5.9 | **10.6** |
| ArXiv (1 month vs 1 month) | TPR@1%FPR | 2.5 | **2.7** |
| Multi-Webdata | TPR@1%FPR | 40.3 | **93.0** |
| LAION-MI | TPR@1%FPR | 2.5 | **8.9** |
| Gutenberg | TPR@1%FPR | 18.8 | **55.1** |

Current MI attacks for foundation models thus cannot be relied on, as we cannot rule out that they are (poorly) inferring membership based on data *features*, without extracting any actual membership leakage from the model. In addition, studies that rely on the MI evaluations on these datasets, also cannot be trusted. For example, the authors of (Panaitescu-Liess et al., 2024) investigate the effect of watermarking on MI attacks and evaluate it on datasets that we show to have clear distribution shifts. Future MI attacks should be evaluated on models with a clear train-test split, e.g., using the Pile (Gao et al., 2020) or a random subset of DataComp (Gadre et al., 2024) or DataComp-LM (Li et al., 2024).

## 2 BACKGROUND AND RELATED WORK

**Web-scale training datasets.** Foundation models are often trained on massive datasets collected from web crawls, such as C4 (Raffel et al., 2020), the Pile (Gao et al., 2020) or LAION (Schuhmann et al., 2022). However, there has been a trend towards using undisclosed training sets for models like GPT-2 to GPT-4 (Radford et al., 2019; OpenAI, 2023), Gemini (Gemini Team, 2023) or DALL-E (Ramesh et al., 2021). Even recent open models like LLaMA (Touvron et al., 2023) do not release information about their training dataset. Some datasets have been released for research purposes, such as RedPajama (Together, 2023), Dolma (Soldaini et al., 2024) or LAION (Schuhmann et al., 2022). Notably, these datasets lack a designated test set.

**Membership inference attacks.** Membership inference attacks aim to determine whether a given data point was used to train a machine learning model (Shokri et al., 2017). Early attacks applied a *global* decision function to all samples (e.g., by thresholding the model's loss (Yeom et al., 2018)). Current state-of-the-art attacks calibrate the attack threshold to the difficulty of each sample (Carlini et al., 2022).

**Membership inference for foundation models.** Membership inference attacks have been applied to LLMs (Shi et al., 2023; Duan et al., 2024; Zhang et al., 2024a; Meeus et al., 2023), diffusion models (Dubiński et al., 2024), CLIP (Ko et al., 2023), and other foundation models. The motivations for these attacks include using them as a component of a privacy attack (Carlini et al., 2021), for evaluating unlearning methods (Shi et al., 2023), or to detect the use of copyrighted data (Meeus et al., 2024b). Due to the lack of a dedicated test set (and possibly even an unknown training set) for the targeted models, many of these works design custom evaluation datasets for membership inference attacks, by collecting sources of data that were likely used (respectively not used) for training.

**Evaluating membership inference.** Membership inference attacks were originally evaluated with average-case metrics such as the ROC AUC on a balanced set of members and non-members (Shokri et al., 2017). More recent work advocates for evaluating the attack's performance in the *worst-case*, typically by reporting the true-positive rate (TPR) at low false-positive rates (FPR) (Carlini et al., 2022).

Membership inference evaluations are usually set up so that a baseline attack (which does not query the target model) achieves 50% AUC (or equal TPR and FPR). Some authors have also considered cases where the attacker has a non-uniform prior (Jayaraman et al., 2020). In either case, the goal of an MI attack is to extract information from the model to beat a baseline inference that does not have access to the model.

Many MI evaluation datasets for foundation models introduce distribution shifts between members and non-members, which allows for baseline attacks with non-trivial success. In concurrent work, Duan et al. (2024) and Maini et al. (2024) identify a temporal shift between members and non-members in one dataset—WikiMIA (Shi et al., 2023)—and demonstrate that some attacks fail when temporal shifts are removed. Our work shows that this issue is much broader: virtually all evaluation sets proposed for membership inference on foundation models are flawed. We further show that existing attacks do not just exploit distribution shifts, but they do so *sub-optimally* and are easily beaten by blind baselines. These attacks thus perform "worse than chance".

## 3 BLINDLY INFERRING MEMBERSHIP

### 3.1 DISTRIBUTION SHIFTS IN MI EVALUATION DATASETS

MI evaluations measure the ability to distinguish a model's training set members from non-members. A baseline attack (with no knowledge of the model) should do no better than random guessing—i.e., an AUC of 50% or equal TPR and FPR. For current MI evaluation datasets for foundation models, this is not the case, due to intrinsic differences between members and non-members. We discuss the most common reasons for this discrepancy below.

**Temporal shifts.** MI evaluation sets based on a hard data cutoff for some evolving Web source (e.g., Wikipedia or arXiv) introduce a temporal shift between members and non-members. We can thus blindly distinguish members from non-members if we can answer a question of the form: "is this data from before or after 2023?"

**Biases in data replication.** Even if we know how the training set was sampled, building an indistinguishable dataset of non-members is challenging (Recht et al., 2019). Slight variations in the procedures used to create the two datasets (Engstrom et al., 2020) could be exploited by a blind attack.

**Distinguishable tails.** Some works filter and process the data to maximally align the member and non-member distributions (e.g., by matching linguistic characteristics). However, distributions that are hard to distinguish *on average* may still be easy to distinguish for *data outliers*.

### 3.2 BLIND ATTACK TECHNIQUES

In this section, we introduce simple "blind" attack techniques to distinguish members from non-members in MI evaluations. The datasets we consider consist of either text, or images and text. For simplicity, we focus only on the text modality in both cases. We do not aim for our blind attacks to be optimal. We prioritize simple and interpretable methods to show that existing evaluation datasets suffer from large distribution shifts. Our blind attacks typically aim to achieve a high TPR at a low FPR (i.e., very confident predictions of membership for as many samples as possible).

**Date detection.** Some text samples (e.g., from Wikipedia, arXiv, etc.) might contain specific dates. Heuristically, it is unlikely that a text contains specific dates referencing the future. Thus, to place an upper-bound on the date at which a text was written, we simply extract all dates present in the text (using simple regular expressions). We then predict that a sample is a member if all referenced dates fall before some threshold. Such an approach can have false positives when a text sample only references dates far in the past.

**Bag-of-words classification.** Explicit dates are just one textual feature we can use. More generally, we can aim to predict membership from arbitrary words in the text. To this end, we train a simple bag-of-words classifier. Here we have to guard against overfitting, since we can always find

Table 2: **Extended version of Table 1, with the results from our blind attacks compared to the best reported MI attack on each evaluation set.** We report the same metrics as used in prior work, and add TPR at 1% FPR if no prior results are reported.

| Section | MI Dataset | Metric | Best Attack | (%) | Ours | (%) |
|---|---|---|---|---|---|---|
| | | *Temporal shifts* | | | | |
| 4.1.1 | WikiMIA | TPR@5%FPR | (Zhang et al., 2024a) | 43.2 | Bag of Words | **94.7** |
| | | AUC ROC | (Zhang et al., 2024a) | 83.9 | Bag of Words | **99.0** |
| 4.1.2 | BookMIA | TPR@5%FPR | (Zhang et al., 2024b) | 33.6 | Bag of Words | **64.5** |
| | | AUC ROC | (Shi et al., 2023) | 88.0 | Bag of Words | **91.4** |
| 4.1.3 | Temporal Wiki | TPR@1%FPR | - | - | Greedy | **36.5** |
| | | AUC ROC | (Duan et al., 2024) | 79.6 | Greedy | **79.9** |
| 4.1.3 | Temporal Arxiv | TPR@1%FPR | - | - | Bag of Words | **9.1** |
| | | AUC ROC | (Duan et al., 2024) | 74.5 | Bag of Words | **75.3** |
| 4.1.4 | Arxiv (all vs 1 mo) | TPR@1%FPR | (Meeus et al., 2023) | 5.9 | Date Detection | **10.6** |
| | | AUC ROC | (Meeus et al., 2023) | 67.8 | Date Detection | **72.3** |
| 4.1.5 | Arxiv (1 mo vs 1 mo) | TPR@1%FPR | (Meeus et al., 2024a) | 2.5 | Greedy | **2.7** |
| | | *Biased replication* | | | | |
| 4.2.1 | Multi-Web | TPR@1%FPR | (Ko et al., 2023) | 40.3 | Greedy | **93.0** |
| | | AUC ROC | (Ko et al., 2023) | 81.7 | Bag of Words | **98.0** |
| 4.2.2 | LAION-MI | TPR@1%FPR | (Dubiński et al., 2024) | 2.5 | Greedy | **8.9** |
| 4.2.3 | Gutenberg | TPR@1%FPR | (Meeus et al., 2023) | 18.8 | Greedy | **55.1** |
| | | AUC ROC | (Meeus et al., 2023) | 85.6 | Bag of Words | **96.1** |

some word combinations that appear in members and not in non-members (e.g., the exact text of the member samples). So we train our classifier on 80% of the members and non-members, and then evaluate the blind attack on the remaining 20%. As some of the evaluation datasets are small, we aggregate results over a 10-fold cross-validation.

**Greedy rare word selection.** The above classifier should work well on average, but might not be optimal at low FPRs. Here, we take a simple greedy approach: we extract all n-grams (for $n \in [1, 5]$) and sort them by their TPR-to-FPR ratio on part of the data (i.e., for each n-gram, we compute the fraction of members and non-members that contain this n-gram). We then pick the n-gram with the best ratio, and repeat this procedure. Given a set of selected n-grams, we evaluate our attack on a held-out set by predicting that a sample is a member if it contains any of these n-grams.

## 4 CASE STUDIES

We now study 9 membership inference datasets proposed for large language models and diffusion models. Table 2 summarizes our results: for each dataset, we create blind attacks that outperform existing MI attacks that have access to a trained model. We average the results from multiple runs and report for the same metric used in prior work, either AUC ROC or TPR at low FPR. Whenever possible, we use the exact datasets released by the authors to ensure that no biases are introduced. For datasets that are not publicly available (arXiv-1 month and Gutenberg (Meeus et al., 2023)), we follow the specific collection steps outlined in the respective work to create similar datasets.

### 4.1 TEMPORAL SHIFTS

#### 4.1.1 WIKIMIA

**The dataset and evaluation.** The WIKIMIA dataset (Shi et al., 2023) selects members from Wikipedia event pages from before 01/01/2017 and non-members from after 01/01/2023. It thus serves as an MI evaluation set for any LLM trained in between those dates. The best reported prior MI attack on this evaluation set is from the Min-K%++ method of Zhang et al. (2024a).

**Our attack.** We first apply our naive date detection attack. We extract all dates in the snippet and check if the latest one is from before 2023. Using this, we obtain a 52.3% TPR at 5% FPR. This already beats the state-of-the-art MI attack evaluated on this dataset (see Table 2). To obtain an even stronger blind attack, we train a bag-of-words classifier on 80% of the dataset, and evaluate on the remaining 20%. This classifier achieves a near perfect TPR of 94.7%. (See Appendix 1 for a visualization of the clear distribution shift in the WikiMIA dataset).

#### 4.1.2 BOOKMIA

**The dataset and evaluation.** This dataset (Shi et al., 2023) is constructed from 512-token length text snippets from various books. Members are selected from books in the Books3 corpus that have been shown to be memorized by GPT-3. Non-members are taken from books that were first published in or after 2023. In the evaluation of Shi et al. (2023), their Min-K% method obtains the highest AUC against GPT-3.

(Duarte et al., 2024) extended this dataset to propose the BookTection dataset. Based on their construction, we expected this dataset to suffer from the same drawbacks as BookMIA and thus do not include it in our study.

**Our attack.** We train a a bag-of-words classifier on 80% of the dataset, and evaluate on the remaining 20%. As shown in Table 2, a bag-of-words classifier achieves a 91.4% AUC ROC and beats state-of-the-art MI attacks against GPT-3.[1]

#### 4.1.3 TEMPORAL WIKI & ARXIV

**The datasets and evaluations.** Duan et al. (2024) hypothesize that some MI attacks evaluated on Wiki-MIA may inadvertently exploit temporal shifts between members and non-members. To showcase this, they create datasets that sample members from the Wikipedia and arXiv snippets in the Pile (Gao et al., 2020) and non-members from the same sources at a later date.[2]

In more detail, in the Temporal arXiv Dataset, members are snippets from arXiv papers posted prior to July 2020 and contained in the Pile training data, while the non-members are sampled from arXiv papers from successively different time ranges after the Pile cutoff of July 2020. The Temporal Wikipedia dataset is also constructed based on the same principle as the WikiMIA dataset, but selects non-member articles from RealTimeData WikiText data created after August 2023 while members are from The Pile from before 03/01/2020. The authors show that existing MI attacks applied to Pythia models improve as the temporal shift increases, with the MI attack of Carlini et al. (2022) performing best.

**Our attacks.** We show that even if existing attacks do rely on some temporal features in the Temporal Wikipedia and Temporal arXiv datasets, they do so sub-optimally. We train a bag-of-words classifier which outperforms prior MI results on these datasets. Specifically, we focus on the "2020-08" split (where non-members are taken from arXiv articles in August 2020, right after the Pile cutoff). There, our blind attack slightly outperforms the AUC and TPR at 1% FPR of the best reported MI attack on both datasets (see Table 2).

---

[1]To control for book- and author-specific features (such as character names), we repeat the experiment with no author appearing in both our classifier's train and test sets. This still results in a significantly above chance AUC ROC of 80.3%.

[2](Duarte et al., 2024) propose a similar ArxivTection dataset constructed using the same temporal principle.

### 4.1.4 ARXIV (ALL VS ONE MONTH)

**The dataset and evaluation.** Meeus et al. (2023) also note that naively re-collecting data from arXiv causes a large temporal shift. Instead, they thus pick the non-members as close after the model's cutoff date as possible. They build an MI evaluation set by taking all arXiv articles from the RedPajama dataset (Together, 2023) as members, and all articles from March 2023 (right after the dataset's cutoff date) as non-members. In contrast to the Temporal arXiv dataset we looked at previously, this dataset uses full arXiv articles rather than just snippets. The authors propose a new MI attack that relies on textual feature extractors, and evaluate it on the OpenLLaMA model (Geng & Liu, 2023) which was trained on RedPajama.

**Our attack.** As this dataset is not public, we replicate a similar setup by taking all articles before March 2023 in RedPajama as the members, and all articles from March 2023 as the non-members. The issue with this dataset is that the distributions of members and non-members are still incredibly easy to distinguish at one end: papers that are *much older* than the cutoff date are guaranteed to be members. And since this dataset uses the full LaTex body of each article, determining a paper's approximate date is very easy: we just look at the paper's citations. We use a regex to find all \cite commands, and extract the year in the citation keyword if it exists. We guess that a paper is a member if it only cites papers from before 2022. This trivial baseline yields $10.6\%$ TPR without any false positives, over twice as high as the TPR at $1\%$ FPR obtained by the best MI attack in (Meeus et al., 2023).

### 4.1.5 ARXIV (ONE MONTH VS ONE MONTH)

**The dataset and evaluation.** In concurrent work, Meeus et al. (2024a) also note the issue of a temporal shift in their previous dataset we discussed above. To remedy this, they build a new MI evaluation set where the non-members are still arXiv articles from March 2023, but the members are now limited to articles from February 2023 (i.e., right before the cutoff date of the RedPajama dataset (Together, 2023)). Unfortunately, this dataset is hence suitable for evaluating MI attacks on a very narrow subset of data. The hope is that this aggressive filtering can remove the temporal shift altogether. And indeed, existing MI attacks fare poorly on this split of members and non-members: the authors show that their best MI attack achieves a TPR of $2.5\%$ at a FPR of $1\%$.

**Our attack.** Even a one month gap could result in a rather significant distribution shift if some research topics are emerging or disappearing from the mainstream. As an illustrative example, if we wanted to show *non*-membership, some topics provide a very strong signal: we find that articles from March 2023 are about $25\times$ more likely to contain the terms "GPT-4" or "iccvfinalcopy" than articles from February 2023.[3]

Biases in the opposite direction are not as clear-cut, but still exist. Using $80\%$ of the data, we greedily select individual words that have the highest TPR-to-FPR ratio on this data (top examples include "pre-registered", "speaker-dependent" or "MMoE"). We then predict that a sample from the held-out set is a member if and only if it contains one of these words. This yields a blind TPR of $2.7\%$ at an FPR of $1\%$, slightly better than the best MI attack that has access to a trained model. While this TPR is small, it does show that temporal biases are persistent even in simple word choices.

## 4.2 BIASES IN DATA REPLICATION AND DISTINGUISHABLE TAILS

### 4.2.1 MULTI-WEBDATA

**The dataset and evaluation.** To evaluate MI attacks on diffusion models, Ko et al. (2023) collect multiple web-scale datasets of captioned images: CC3M (Sharma et al., 2018), CC12M (Changpinyo et al., 2021), (LAION (Schuhmann et al., 2022) and MS-COCO (Lin et al., 2014)). They argue that since these datasets are sampled from a common global source (i.e., captioned images from the Web), they can serve as a natural evaluation set for MI attacks. Specifically, the authors train a diffusion model on CC12M (the members), and sample the non-members from the union of

---

[3]GPT-4 was released on March 14th 2023. The submission deadline for ICCV 2023 was on March 8th 2023.

the other three datasets.[4] Ko et al. (2023) evaluate the performance of three newly proposed MI attacks on target models from the ViT (Dosovitskiy et al., 2020) and ResNet (He et al., 2016) families trained on CC12M.

**Our attack.** For simplicity, we focus only on the image captions and ignore the visual content. Using our greedy n-gram selection method on the captions, we achieve a TPR of $93\%$ at a $1\%$ FPR, far superior to the best MI attack methods evaluated on these datasets (see Table 2). It is likely that incorporating visual features into the classifier could lead to an even stronger blind attack.

### 4.2.2 LAION-MI

**The dataset.** Dubiński et al. (2024) explicitly try to control for and minimize biases in data replication. However, their techniques still leave distinguishable tails. To create an evaluation dataset for a model trained on LAION (Schuhmann et al., 2022), they rely on the multilingual-LAION dataset (Schuhmann et al., 2022) to sample non-members. This dataset was sampled from the Web in a similar way to LAION, except that the captions are not in English and the images were not filtered for quality.

To align the two distributions, the authors first apply the same selection process that was used to select images in LAION: (1) they discard all images from multilingual LAION with an "aesthetics score" below some threshold; (2) they translate the non-member captions to English, using Facebook's M2M100 1.2B. They then train a distinguisher on the image captions, and select a subset of the data where members and non-members are hard to distinguish *on average*. The authors use a PCA visualization to confirm that the two distributions closely match.

The authors evaluate multiple MI attacks in white-box, grey-box and black-box settings, obtaining a best score of 2.5% TPR@1%FPR on their LAION-MI dataset.

**Our attack.** We posit that *outlier* members would still be easy to distinguish from non-members. To provide some intuition, we started by looking for individual *characters* that appear much more frequently in members than in non-members. We show the five characters with the highest distinguishing power in Table 3.

We note that there are non-English characters (Russian 'о' and 'т') that appear predominantly in *members*. This is likely because the original LAION captions were selected on the basis of being *primarily* in English. In contrast, since the captions from the non-members were output by a translation model, they are unlikely to contain non-English characters. The presence of special characters such as \xa0 (a non-breaking space) are similarly due to biases in translation: this character is not part of the translation model's output vocabulary.

Table 3: **Characters with the highest distinguishing power among members and non-members for LAION-MI.**

| Character | Members (%) | Non-members (%) |
|---|---|---|
| \| | 0.60 | 0.00 |
| \xa0 | 0.30 | 0.00 |
| ... | 0.21 | 0.00 |
| о | 0.23 | 0.03 |
| т | 0.15 | 0.03 |

To boost the power of this approach, we apply our greedy selection algorithms to all n-grams of up to 5 characters. We use $80\%$ of the data to pick the n-grams with the highest TPR-to-FPR ratio until we hit a $1\%$ FPR. At that point, we obtain an $8.9\%$ TPR on the evaluation set—surpassing the best MI attack.

---

[4]Wu et al. (2023) use a similar approach, and choose members and non-members from different text-vision datasets.

### 4.2.3 PROJECT GUTENBERG

**The dataset and evaluation.**   This dataset (Meeus et al., 2023) consists of books from Project Gutenberg. The members are books contained in the RedPajama dataset (Together, 2023), more specifically the set of "PG-19" books collected by Rae et al. (2019) which were all originally published before 1919. The non-members consist of books added to the Project Gutenberg repository after the creation of the PG-19 corpus in February 2019. To mitigate the obvious distribution shift in the publication years of members and non-members, Meeus et al. (2023) filter both the members and non-members to only contain books published between 1850 and 1910 (i.e., members are books published between 1850 and 1910, which were added to Project Gutenberg before February 2019. Non-members are books that were originally published in the same period, but added to Project Gutenberg after February 2019). The authors evaluate a new MI attack (see Section 4.1.4 for details) against the OpenLLaMA model (Geng & Liu, 2023) trained on RedPajama.[6]

**Our attack.**   While the publication dates of the members and non-members are similarly distributed, we find that *the date on which a book is added to Project Gutenberg* still introduces a noticeable distribution shift. Indeed, it appears that the format of the preface metadata that Project Gutenberg adds to books has changed at some point between the collection of PG-19 and the non-member corpus (e.g., the link to the HTML version of a book used to have a `.htm` file extension, and this was changed to a `.html` extension for books added after a certain date). By exploiting such small discrepancies, our greedy n-gram attack applied to the first 1,000 characters of a book achieves 55.1% TPR at a 1% FPR, 3 times higher than the best MI attack evaluated on this dataset.

Of course, it may be possible to preprocess this particular dataset to get rid of these formatting discrepancies. Nevertheless, we believe our results illustrate how brittle the process of re-collecting an identically distributed non-member set can be. Moreover, even if we ignore any formatting information, and apply our attack directly to the book text, we can still distinguish members and non-members far better than chance, with a TPR of 16.6% at a 1% FPR.

## 5   A PATH FORWARD: THE PILE, DATACOMP AND DATACOMP-LM

As we have seen, building MI evaluation datasets *a posteriori* is incredibly challenging, due to the many possibilities for distribution shift. It may be possible to apply more rigorous filtering techniques to minimize these shifts, so that even strong blind attacks cannot reliably distinguish members and non-members. Any MI attack's performance should then be compared against these strong blind baselines.

An alternative avenue is to forego MI evaluations on arbitrary foundation models, and focus on those models for which identically distributed sets of members and non-members are available. A prime example is the Pile (Gao et al., 2020), which has an official test set. Models trained on the Pile training set—e.g., Pythia (Biderman et al., 2023) or GPT-NeoX-20B (Black et al., 2022)—are thus a popular target for MI attack evaluations (see e.g., (Duan et al., 2024; Maini et al., 2024; Li et al., 2023)). Unfortunately, the Pile only contains text, and is too small to train state-of-the-art language models.

A more recent and broadly applicable alternative is to use models trained on the DataComp (Gadre et al., 2024) and DataComp-LM (Li et al., 2024) benchmarks. These benchmarks contribute multiple models (ViTs and LLMs) trained on data *sampled randomly* from a larger dataset. More precisely, these benchmarks introduce two datasets, CommonPool and DCLM-POOL, which contain respectively 12.8 billion image-text pairs and 240 trillion text tokens sampled from Common Crawl. To enable experiments at smaller budgets, the authors produce smaller data pools sampled randomly from the full pool. They then train a variety of models on each pool (see Table 4).[5,6]

A minor complication is that most models in the DataComp and DataComp-LM benchmarks are not trained on the entirety of a pool. Rather, the benchmarks encourage models to be trained on *filtered* datasets: given a pool of data $\mathcal{P}$ and a binary filter $f : \mathcal{X} \mapsto \{0, 1\}$, models are trained on the filtered

---

[5]https://github.com/mlfoundations/datacomp
[6]https://github.com/mlfoundations/dclm.

Table 4: **Models trained on datasets with *random* train-test splits for DataComp (Gadre et al., 2024) and DataComp-LM (Li et al., 2024)**. For each dataset pool, we report the pool's absolute size (in image-text pairs, respectively tokens) and its relative size compared to the global pool that it was randomly sampled from. Gadre et al. (2024) and Li et al. (2024) have pretrained ViTs and LLMs on each pool size.

| Modality | Pool Name | Pool Size | Models |
|---|---|---|---|
| Vision+Text | CommonPool Large | 1.280B (10%) | ViT-B/16 |
| | CommonPool Medium | 0.128B (1%) | ViT-B/32 |
| | CommonPool Small | 0.013B (0.1%) | ViT-B/32 |
| Text | DCLM 400M-1x | 0.47T (0.2%) | 412M LLM |
| | DCLM 1B-1x | 1.64T (0.2%) | 1.4B LLM |
| | DCLM 1B-5x | 8.20T (3.4%) | 1.4B LLM |
| | DCLM 7B-1x | 7.85T (3.3%) | 6.9B LLM |
| | DCLM 7B-5x | 15.7T (6.5%) | 6.9B LLM |

pool $\{x \in \mathcal{P} \mid f(x)\}$. This training set is thus no longer a random subset of the full dataset. But this is easily remedied by applying exactly the same filter $f$ to select the non-members.

We encourage future work on MI attacks for foundation models to evaluate their methods on these models and datasets.

As part of their experiments, Gadre et al. (2024) and Li et al. (2024) trained CLIP models of various sizes and a 1B parameter language model on *random* subsets of these datasets. These models and datasets might thus form an ideal testbed for MI evaluations on web-scale text and vision models.

Future web-scale dataset and model developers could similarly release an official IID (independent and identically distributed) split to enable rigorous MI evaluations for a wider variety of models and modalities.

## 6 CONCLUSION

We have shown that current evaluations of MI attacks for foundation models cannot be trusted, as members and non-members can be reliably distinguished by simple blind attacks with no knowledge of the model. State-of-the-art MI attacks may thus be ineffective at extracting any actual membership information, and cannot be relied on for applications such as copyright detection or auditing unlearning methods. The MI evaluation datasets we investigated should likely be discarded due to the large distribution shifts between members and non-members, and replaced by datasets with a random train-test split—e.g., the Pile or DataComp.

## 7 REPRODUCIBILITY

Our results can be reproduced using the code-base provided in the supplementary material. The code-base includes most of the datasets, and scripts to download the datasets that were too large to include.

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

## A  APPENDIX

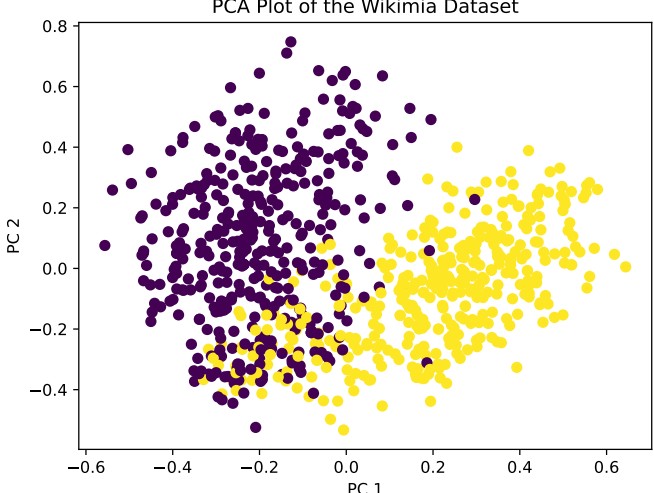

Figure 1: In some cases, the distribution shift is easy to visualise. Here is the PCA plot of the WikiMIA dataset which shows members and non-members forming distinguishable clusters.

