# OpenReview forum: "Blind Baselines Beat Membership Inference Attacks for Foundation Models"
_ICLR.cc/2025/Conference — Submitted to ICLR 2025_

### Official Review · Reviewer_o2wH · 2024-10-26

**Soundness:** 2
**Presentation:** 3
**Contribution:** 2
**Rating:** 3
**Confidence:** 5

**Summary:**

The paper argues that previous evaluations of membership inference attacks are flawed due to the distributional differences between members and non-members. The paper analyzes nine datasets and demonstrates that blind attack techniques, such as date detection (assume some text samples contain dates), bag-of-words classification (classifier is trained on 80% of the members and non-members, then test on the left 20% members), and greedy rare word selection, outperform previous state-of-the-art (SOTA) methods in the evaluation metric.

**Strengths:**

1. The paper provides extensive experimental results to support their claims. These results demonstrate that blind attack techniques outperform state-of-the-art methods under their settings.

**Weaknesses:**

1. The assumption of this paper on "blind" is not correct. "Blind" should be on both the model and data set [2021], but this paper relies on too much target dataset information. For example, one of the proposed methods only works if the dataset contains data information, and another method even needs 80% of labeled member data as the attacker's training samples. From my understanding of the literature, this rich information may not be available to other membership inference attacks, potentially giving the proposed blind attack an unfair advantage.

Hui, Bo, Yuchen Yang, Haolin Yuan, Philippe Burlina, Neil Zhenqiang Gong, and Yinzhi Cao. "Practical blind membership inference attack via differential comparisons.", 2021

2. The paper provides limited experimental details. For instance, it does not specify which models were targeted for the membership inference attacks.

3. The paper proposes ideas for constructing better datasets for evaluating membership inference attacks, but it does not provide experimental results or analysis on whether the blind attack would still outperform SOTA methods on these improved datasets.

4. Current membership inference attacks are typically evaluated across multiple datasets. For example, Zhang et al. [2024a] evaluate their Min-K%++ attack on Wikipedia, GitHub, Pile CC, PubMed Central, and many other datasets to demonstrate generalizability. However, the blind attack’s performance on other datasets is not explored in the paper, making it difficult to conclude that current evaluations are entirely flawed based on the results from just one dataset.

Jingyang Zhang, Jingwei Sun, Eric Yeats, Yang Ouyang, Martin Kuo, Jianyi Zhang, Hao Yang, and Hai Li. Min-K%++: Improved baseline for detecting pre-training data from large language models. arXiv preprint arXiv:2404.02936, 2024a.

**Questions:**

Duan et al. (2024) propose that temporal shifts can influence the performance of membership inference attacks. While you mention that there are differences between your paper and this concurrent work, from my perspective, both papers seem to demonstrate similar findings. Could you elaborate further on the differences between your work and Duan et al. [2024]?

Michael Duan, Anshuman Suri, Niloofar Mireshghallah, Sewon Min, Weijia Shi, Luke Zettlemoyer, Yulia Tsvetkov, Yejin Choi, David Evans, and Hannaneh Hajishirzi. Do membership inference attacks work on large language models? arXiv preprint arXiv:2402.07841, 2024.

---

> ### Author Response · Authors · 2024-11-18
> **Response**
>
> Thank you for your detailed feedback and for taking the time to review our work.
> Unfortunately, it seems that the reviewer has misunderstood the core contribution of our work. We hope the response below can clarify what our work does.
>
> ---
>
> >The assumption of this paper on "blind" is not correct. "Blind" should be on both the model and data set [2021], but this paper relies on too much target dataset information.
>
> The referenced paper uses “blind” in a different sense than ours. In Hui et al., a blind attack is one that only has black-box access to the model.
>
> We consider attacks that do not have any access to the model at all! Such an attack shouldn’t even be able to do membership inference since it can’t possibly infer any information from the model. So in principle it should be irrelevant how much information our attacks have about the target dataset. And yet we show that such “blind” attacks can still distinguish “members” and “non-members” because the evaluation datasets are badly constructed.
>
> We will clarify this terminology in our paper.
>
> >The paper provides limited experimental details. For instance, it does not specify which models were targeted for the membership inference attacks.
>
> This is a misunderstanding of our paper. Our attacks do not use any access to a model—this is a core point of our entire work.
>
> >The paper proposes ideas for constructing better datasets for evaluating membership inference attacks, but it does not provide experimental results or analysis on whether the blind attack would still outperform SOTA methods on these improved datasets.
>
> By definition, a dataset with an IID split would not allow any blind attack to work better than random chance.
> For completeness, we verified this on the PILE, and found that our blind attacks indeed cannot distinguish the train and test sets better than random.
>
> >Current membership inference attacks are typically evaluated across multiple datasets. For example, Zhang et al. [2024a] evaluate their Min-K%++ attack on Wikipedia, GitHub, Pile CC, PubMed Central, and many other datasets to demonstrate generalizability. However, the blind attack’s performance on other datasets is not explored in the paper, making it difficult to conclude that current evaluations are entirely flawed based on the results from just one dataset.
>
> Indeed, we are not claiming that existing evaluations are completely flawed. We are just saying that the use of the specific datasets we consider should be avoided because these datasets do not provide any meaningful signal.
>
> So, for example, if a paper evaluated their MI attack on WikiMIA and the Pile, we argue it would be better if they just evaluated on the Pile.
>
> Also note that we are not claiming that any individual attack (such as Min-K%++) is flawed. We are saying that (parts of) the evaluations of these attacks are flawed. It is entirely possible that the attacks are very effective, but the current evaluations cannot be trusted to show this.
>
> >Duan et al. (2024) propose that temporal shifts can influence the performance of membership inference attacks. Could you elaborate further on the differences between your work and Duan et al. [2024]?
>
> Past works have indeed pointed out issues of distribution shifts for MI evaluations. However, these works have all focused on one specific type of shift present in one or two datasets (temporal wiki and arxiv).
>
> We go further and present a systematic analysis of biased members and non-members across 9 published MI evaluation datasets of three different types.  We show with experiments that the shift is so severe, that any MI evaluation scores on these datasets cannot be trusted. We also show, more alarmingly, that this issue pervades other kinds of datasets - specifically our other two types of dataset constructions (biases in data replication, distinguishable tails) that are not constructed based on a temporal split.
>
> We will clarify our contributions and relationship to prior work.

---

> > ### Comment · Reviewer_o2wH · 2024-11-24
> >
> > Thank you for your response. I now understand that your work aims to highlight the shortcomings of existing datasets used for MI attacks in text and demonstrate that a very naive approach based on text features can achieve strong performance. I agree this is an important issue, as noted by the other reviewers. However, I don’t believe it is accurate to describe your approach as "blind," as its success relies on knowing the distribution shift between training and testing data and then selecting or training a classifier accordingly.
> >
> > I have also reviewed the comments from the other reviewers and I have a similar feeling. Therefore, I will maintain my score.

---

### Official Review · Reviewer_zRbN · 2024-10-28

**Soundness:** 3
**Presentation:** 3
**Contribution:** 2
**Rating:** 5
**Confidence:** 4

**Summary:**

This paper reveals that current evaluations of membership inference (MI) attacks on foundational models are flawed due to the use of different distributions when sampling members and non-members. The authors demonstrate this issue through an analysis of nine published MI evaluation datasets. They show that directly classify the samples in the MI evaluation datasets can outperform existing MI attacks. This finding indicates that current evaluation methods cannot accurately reflect the membership leakage of a foundational model's training data. This paper also proposes simple blind attack techniques, such as date detection and bag-of-words classifiers, which remain effective on datasets designed to eliminate distribution differences between members and non-members. The authors suggest that future MI attack evaluations should be conducted on models with a clear train-test split.

**Strengths:**

- This paper focuses on the irrationality of MI evaluation datasets is important, especially in an era where foundation models are widely applied.
- This paper analyzes 9 published MI evaluation datasets, demonstrating that blind attacks outperform existing MI attacks on these datasets. This reveals the incompleteness of current MI evaluations.
- The attack methods proposed in this paper perform exceptionally well, showing significant performance improvements compared to existing MI attacks.

**Weaknesses:**

- The comparison experiment setup is unclear. Were the same data conditions used the experiment section? (see Q1)
- The core of this paper is to point out the shortages of existing MI attacks on foundation models. However, in the introduction, the discussion does not revolve around this point but rather focuses on how simple attacks can also achieve good results. It is recommended to revise the structure of the introduction to highlight the main contributions of the paper.
- The experimental section is divided into sections based on the datasets, which makes it difficult to correspond with the previously mentioned common reasons for the intrinsic differences. This hinders the reader's understanding of the experiments and the paper's arguments.

**Questions:**

Q1: In Section 3, the authors first extract all dates present in the text and then proceed with date detection attacks. Are samples lacking dates excluded from your inference attacks? Specifically, in the experimental section, are the 'Ours' attack results shown in Table 2 based on a dataset that has filtered out samples without dates? Similarly, are the 'Best Attack' results measured on such a filtered dataset, or are they directly taken from the original papers? This distinction is crucial as it determines the fairness of your comparison experiments.

Q2: In your date detection attack, could you elaborate on how the date threshold is selected? The relevant section does not detail the method for choosing the threshold. Is it a matter of testing various thresholds and selecting the one that yields the best attack performance?

Q3: Regarding your Bag-of-words classification attack, what is the underlying insight or motivation for this approach? Why can different word combinations be used to infer membership properties, particularly without any interaction with the foundational models?

Q4: This paper primarily concentrates on the datasets used for MI evaluation, but it does not account for the influence of foundational models. However, the objective of MI attacks is to discern differences between a model's behavior on members and non-members. Even with datasets that are easily distinguishable based on dates or words, existing attacks still fail to achieve satisfactory results after considering the foundational models. Does this suggest that the current use of these datasets remains valid? Furthermore, by focusing on direct distinctions in membership labels and disregarding interactions with foundational models, are the signals you use for differentiation, such as dates and words, being utilized by existing MI attacks that do interact with foundational models? If current MI attacks are not leveraging your information, does it further indicate that the use of these datasets is reasonable (at least at current stage)?

Q5: In Section 3.1, this paper proposes three common reasons for the intrinsic differences between member and non-member samples. However, since different MI evaluation datasets are constructed using various strategies, it is unclear which aspects are evaluated with respect to these datasets. It would be beneficial to clarify this point in Table 2.

---

> ### Author Response · Authors · 2024-11-18
> **Response**
>
> Thank you for your detailed feedback and for taking the time to review our work. We understand that the significance of our findings may not have been fully apparent, and we appreciate the opportunity to clarify the key contributions of our paper.
>
> ---
>
>
> >The comparison experiment setup is unclear.  In Section 3, the authors first extract all dates present in the text and then proceed with date detection attacks. Are samples lacking dates excluded from your inference attacks?
>
> For samples that do not contain a date, our attack simply has to revert to random guessing (or, to minimize false positives, we simply output “not member”). So we don’t do any special filtering here. We apply our attack (and prior ones) to the full dataset.
>
> >In your date detection attack, could you elaborate on how the date threshold is selected?
>
> Since we know the dataset and how it was constructed, we simply select the dataset’s cutoff date as our threshold date. More generally, we could also iterate over a few choices of dates and choose the best one based on a validation set.
>
> >The core of this paper is to point out the shortages of existing MI attacks on foundation models. However, in the introduction, the discussion does not revolve around this point but rather focuses on how simple attacks can also achieve good results.
>
> Our point in the introduction was to say that since blind (“naive”) attacks can achieve high scores, existing MI evaluations are flawed. We will clarify this.
>
> >The experimental section is divided into sections based on the datasets, which makes it difficult to correspond with the previously mentioned common reasons for the intrinsic differences.
>
> We do break down the evaluation section into subheadings 4.1, 4.2, and 4.3, each corresponding to a particular reason for differences as the reviewer suggests. Within each of these subsections, we have one sub-subsection per defense.
> Does the reviewer have a different structure in mind that would be easier to follow?
>
> >Regarding your Bag-of-words classification attack, what is the underlying insight or motivation for this approach? Why can different word combinations be used to infer membership properties, particularly without any interaction with the foundational models?
>
> The motivation is basically a generalization of our date-extraction attack. Since the members and non-members come from different distributions, there are likely some words that are more likely in one distribution than the other, that can be used to make a good guess.
>
> >This paper primarily concentrates on the datasets used for MI evaluation, but it does not account for the influence of foundational models. However, the objective of MI attacks is to discern differences between a model's behavior on members and non-members.
> Even with datasets that are easily distinguishable based on dates or words, existing attacks still fail to achieve satisfactory results after considering the foundational models. Does this suggest that the current use of these datasets remains valid? Furthermore, by focusing on direct distinctions in membership labels and disregarding interactions with foundational models, are the signals you use for differentiation, such as dates and words, being utilized by existing MI attacks that do interact with foundational models? If current MI attacks are not leveraging your information, does it further indicate that the use of these datasets is reasonable (at least at current stage)?
>
> This is a good question. Current MI attacks indeed perform very poorly even on very biased datasets (worse than our blind baselines). While this suggests there is room for improvement, we would still argue for these datasets to be dropped. Indeed, if a new attack comes along that does much better on these datasets, we wouldn’t know if this is because the attack actually is better at extracting membership signal from the model, or if it is just picking up the biased features that our attacks rely on.

---

> > ### Comment · Reviewer_zRbN · 2024-11-19
> >
> > Thank you for addressing my comments. I agree that for text-based foundation models, there will indeed be different lexical combinations and data distributions before and after certain dates. However, for vision foundation models, the distribution of training and testing data typically does not exhibit such a significant gap. Therefore, would blind attack not be generalizable to other domains and has limited extension ability?
> >
> > Furthermore, I think that merely highlighting the shortcomings of existing benchmarks is insufficient. The MIA proposed in this paper merely reveal the distribution differences in the data for foundation models, neglecting the differences that may exist in the model's processing of these data. As the authors mentioned in their response, ''we wouldn't know if this is because the attack actually is better at extracting membership signals from the model, or if it is just picking up the biased features that our attacks rely on''. There lacks an exploration of how foundation models handle data from before and after different dates, even if it were just some analytical experiments. Such exploration could provide the MIA community with new insights or observations, and would make the contribution of this paper more sufficient.

---

> > > ### Author Response · Authors · 2024-11-19
> > > **Response**
> > >
> > > We're glad we could address your main comments.
> > > Regarding your remaining questions:
> > >
> > > > However, for vision foundation models, the distribution of training and testing data typically does not exhibit such a significant gap. Therefore, would blind attack not be generalizable to other domains and has limited extension ability?
> > >
> > > We are confident that similar issues apply to vision foundation models for the following reasons:
> > >
> > > 1. Training sets for modern vision foundation models consist of images *and* text (e.g., LAION, CC, etc). These datasets don't have an official train-test split, and so MI attacks on these models still need to create a set of non-members after the fact. This is exactly what happens in the LAION-MI and Multi-Webdata case-studies in our paper.
> > >
> > > 2. We focused on text-only blind attacks in our paper mainly for simplicity (i.e., we can use simple models like bag of words). But we see no reason why we shouldn't be able to train image models to distinguish between members and non-members for these datasets.
> > >
> > > > There lacks an exploration of how foundation models handle data from before and after different dates, even if it were just some analytical experiments
> > >
> > > It is not clear to us what the point of such an experiment would be. Even if *current* MI attacks use (or don't use) the same features as our blind attacks, this says nothing about future attacks. Our worry is precisely that future attacks would be evaluated on the same biased datasets, and that we then have no idea if the attacks actually work or not.

---

> > > > ### Comment · Reviewer_zRbN · 2024-11-23
> > > >
> > > > Thank you for your response. I understand that the focus of your work is to highlight the inherent shortcomings of the datasets used to evaluate MIA in LLMs, which I agree is a significant challenge in this field. However, for a top-tier conference, simply pointing out issues with a naive attack method may not be enough in terms of contribution and novelty. My suggestion would be to either develop a more accurate method for assessing MIA in LLMs or to conduct a deeper analysis of MIA performance on existing evaluation datasets. Such an analysis is expected to reveal the relationship between current MIA performance and differences in data distribution. Providing solutions or even potential solutions to address this important problem would enhance the significance of your work. Based on these considerations, I will maintain my current score.

---

### Official Review · Reviewer_GGin · 2024-10-31

**Soundness:** 2
**Presentation:** 2
**Contribution:** 2
**Rating:** 5
**Confidence:** 4

**Summary:**

This paper examines the datasets used in evaluating membership inference attacks on large language models and text-to-image generation models. The authors argue that current MIA evaluations are unreliable, as it is possible to differentiate members from non-members through blind attacks that do not utilize any information about the target model. Consequently, they suggest that state-of-the-art MIAs may not actually extract membership information effectively. To improve evaluation, the authors recommend using datasets with minimal distribution shifts between members and non-members, such as Pile or DataComp.

**Strengths:**

- The paper investigates a significant issue in MIA research, highlighting the importance of unbiased evaluation datasets for accurately benchmarking attack effectiveness on text-based large foundation models.

- It provides a systematic evaluation of various datasets and baseline attacks, identifying three common distribution shift patterns that influence the success of MIAs.

**Weaknesses:**

- The authors claim that current state-of-the-art MIAs fail to extract meaningful membership information, relying only on biased dataset evaluation results. However, this assertion may be overstated, as blind attacks use dataset-specific prior information (e.g., timestamps), which the proposed state-of-the-art attacks may intentionally avoid as they may aim to propose a general attack. These attacks might still capture useful membership signals, albeit weaker than the dataset-specific prior information. To better support this claim, experiments on less biased datasets (like Pile or DataComp, as suggested) are necessary. If state-of-the-art methods perform close to random guessing on such datasets, it would indicate their inability to capture membership information effectively.

- As a path forward, the paper advocates for future MIA evaluations using PILE, DataComp, or DataComp-LM. However, it is unclear whether these datasets also suffer from distribution shift issues. A simple approach to evaluate this would be to apply the proposed blind attacks on these datasets; if the success rate is near random guessing, it could indicate that these datasets are indeed less biased by distribution shifts, at least concerning the three identified types of shift.

- As highlighted by Dubinski et al. (2024), different splits of training and evaluation sets can yield significantly varied membership inference attack results. To ensure robustness of the evaluations, it would be beneficial to repeat the experiments with different random dataset splits, recording the mean and variance of attack success rates. This approach would provide a more reliable comparison between blind attacks and existing MIAs.

- The novelty and technical contributions of this paper appear incremental. Distribution shift issues in evaluation datasets have been previously discussed by Duan et al. and Maini et al., and while I appreciate the systematic evaluations in this paper, it largely provides a measurement study rather than new technical contributions or insights. Thus, the paper might lack the innovation typically expected at top-tier conferences, just my two cents.

**Questions:**

- How do state-of-the-art attacks perform relative to blind attacks on unbiased datasets?

- Are the recommended datasets (PILE, DataComp, or DataComp-LM) genuinely unbiased?

- What are the effects of repeating experiments using different dataset splits on the evaluation outcomes?

**Details Of Ethics Concerns:**

No ethical concerns are involved.

---

> ### Author Response · Authors · 2024-11-18
> **Response**
>
> Thank you for your detailed feedback and for taking the time to review our work. We understand that the significance of our findings may not have been fully apparent, and we appreciate the opportunity to clarify the key contributions of our paper.
>
> ---
>
> >The authors claim that current state-of-the-art MIAs fail to extract meaningful membership information, relying only on biased dataset evaluation results. However, this assertion may be overstated
>
> This is not the claim we make.
> We claim that the performance of state-of-the-art MIAs cannot be distinguished (or is even outperformed) by naive baselines without access to the target model.
> Our baselines indeed use information that an MI attack might have a hard time extracting. But the point here is that an attack could, in principle, use such features which give no information about the actual ability to extract membership signal.
>
> >As a path forward, the paper advocates for future MIA evaluations using PILE, DataComp, or DataComp-LM. However, it is unclear whether these datasets also suffer from distribution shift issues.
>
> Since the train and test sets of these datasets were selected IID, we are guaranteed that no distribution shift exists.
> For completeness, we tested our attack strategies on the train/test split of the PILE, and got an advantage over random guessing that was not statistically significant.
>
>
> > To ensure robustness of the evaluations, it would be beneficial to repeat the experiments with different random dataset splits, recording the mean and variance of attack success rates.
>
> The results reported are averaged over repeated experiments (10-fold cross validation as mentioned on line 197) with random splits - in the cases where the blind attacks need a “training” phase - such as the bag of words attack and the greedy rare word selection attack.
>
> Date detection based blind attack does not require any “training data” so the entire dataset is used as the test data. Since there is no randomness in this method: neither in the train-test split nor in the actual attack method, this experiment is not repeated and cross-validated. Any repetition will give the same score.
>
> >The novelty and technical contributions of this paper appear incremental. Distribution shift issues in evaluation datasets have been previously discussed by Duan et al. and Maini et al.
>
> Duan et al. and Maini et al. focus on temporal shifts within a single dataset (WikiMIA), whereas we provide a systematic analysis of biased members and non-members across three types of biases in nine published MI evaluation datasets. In contrast to these works, we also don’t merely show that a distribution shift exists, but that it is large enough to invalidate all MI evaluations conducted on these datasets to date.
>
> We believe this broader perspective deserves greater attention in the community, as we show that issues in MI dataset creation are not a one-off event, but a systematic issue that plagues the entire field. And yet, many of these datasets are still being used for evaluating MI attacks (as a timely example, ref [3] below which was recently awarded an Outstanding Paper Award at EMNLP proposes a new MIA method and evaluates it on WikiMIA and BookMIA—datasets that we show are severely biased).
> So can we truly trust these reported numbers? It is crucial for researchers to approach evaluation with caution to avoid producing and propagating misleading results.
>
> We believe that publishing this work in a venue such as ICLR, which plays a pivotal role in shaping the trajectory of machine learning research, will help ensure these concerns are brought to the forefront of the community's attention.
>
> >How do state-of-the-art attacks perform relative to blind attacks on unbiased datasets?
>
> On a truly unbiased dataset, our blind attacks would perform no better than chance (by definition).
> Prior work (e.g., Duan et al. 2024) show that current MIAs also do essentially random guessing on such datasets, but it is possible that future, stronger attacks could extract meaningful signal from unbiased datasets.
>
> [1] Do Membership Inference Attacks Work on Large Language Models? ICLR 2024.
>
> [2] LLM Dataset Inference: Did you train on my dataset? Arxiv,2406.06443.
>
> [3] Pretraining Data Detection for Large Language Models: A Divergence-based Calibration Method. Arxiv:2409.14781.

---

> > ### Comment · Reviewer_GGin · 2024-11-23
> >
> > Thank you for your response. I have carefully reviewed the rebuttal and the comments from other reviewers. For a top-tier conference, it is important for the paper to demonstrate sufficient scientific contributions. Including potential solutions would be one direction that could significantly strengthen the work. I also recommend that the paper quantitatively incorporate experimental results from prior MIA attacks using datasets such as PILE, DataComp, or DataComp-LM to support its claims better. Based on the above reasons, I will maintain my score.

---

### Official Review · Reviewer_f6sv · 2024-11-03

**Soundness:** 4
**Presentation:** 4
**Contribution:** 1
**Rating:** 5
**Confidence:** 5

**Summary:**

This paper demonstrates that existing MI evaluations for foundation models perform poorly due to distribution shifts between member and non-member data. The authors show that simple "blind" attacks, which do not query the model, can outperform state-of-the-art MI attacks on common MI evaluation datasets. They identify temporal shifts, biases in data replication, and distinguishable tails as common causes for the distribution mismatch between members and non-members.

**Strengths:**

- While this topic has been explored by several recent works (both concurrent and prior), this work goes a step beyond to demonstrate the extend of distributional differences between members and non-members, for both LLM and VLM evaluation data for membership inference.

- The paper is well written and supports most of its claims with empirical evidence and extensive evaluation.

**Weaknesses:**

The submission has significant issues regarding originality and the characterization of related work. The authors' framing of certain works as "concurrent" appears to minimize substantial overlaps, particularly with [1] and [2] which preceded the ICLR deadline by 4 and 8 months respectively. This timeframe makes it difficult to justify as concurrent research. The paper's main conclusion about flawed non-member selection methods introducing detectable distributional shifts largely mirrors the findings already established in [2].

The conclusion (L413) takes a problematic stance by seemingly absolving model trainers of accountability. Instead of abandoning membership inference evaluations, research should focus on developing methods that either avoid non-member requirements (like data-extraction attacks) or leverage trainer-provided evaluation data. Dismissing these evaluations would encourage (proprietary) model trainers to evade scrutiny of their data usage practices.

The claim on L464-465 about "no knowledge of the model" requires clarification. The paper's "blind" baselines actually incorporate significant domain knowledge about data collection patterns (e.g., dates and special tokens). The authors should explicitly state that "blind" specifically refers to lack of model access, as the attacks still utilize direct data knowledge and split information to construct rules and meta-classifiers.

## Other Comments

- L89: Current state of the art is RMIA [3], nor LiRA.

- L89: Please provide some more context for the 'standard' membership inference game (member and non-members should be same distribution etc.) Context is especially important for this work to understand the nuances behind different member/non-member data distributions.

- L101: "Many of these" - Please be exact. Consider adding a table that talks about MI attack attempts on foundation models, the benchmarks they use (propose or new), and whether they suffer from the train-test split issues that the authors mention here.

- L154:  The assumption about future dates is oversimplified and overlooks legitimate cases in fiction, climate research, and policy documents

- Table 1 appears redundant given Table 2's more comprehensive presentation

- Table 2: Please make a distinction between datasets for LLMs and those for VLMs.

- L467: "... auditing unlearning methods" - there are several works describing better ways to audit unlearning [4]. It should also be pointed out that these membership-inference attacks do not work well to begin with even with properly split train/test data [2], so it is not surprising that it will not be used to audit unlearning.

#### References

- [1] Maini, Pratyush, et al. "LLM Dataset Inference: Did you train on my dataset?." arXiv:2406.06443 (2024).
- [2] Duan, Michael, et al. "Do membership inference attacks work on large language models?."COLM, 2024
- [3] Zarifzadeh, Sajjad, Philippe Liu, and Reza Shokri. "Low-Cost High-Power Membership Inference Attacks." ICML, 2024.
- [4] Lynch, Aengus, et al. "Eight methods to evaluate robust unlearning in llms." arXiv:2402.16835 (2024).

**Questions:**

- Some of the "blind" baselines rely on detecting data references like 2024 etc. As an adversary cognizant of the training cutoff (or even an auditor), a simple solution to fix this shift would be arbitrarily replacing all such data references for non-members, maybe shift them by N nears back to match the cutoff range of the model. What happens in such a scenario? Can "blind" attacks still be successful?

- In Table 2, why are some of the entries missing values, or entire metrics (AUC ROC) not reported? These datasets are publicly available (which is how the authors get their attack's results to begin with) so I do not see why corresponding values cannot be filled in for existing works.

- I am confused by the 'Greedy rare word selection' protocol- is the sorting done using the TPR/FPR ratios on test data? If so, one can always design and selective use metrics to get good performance on test data if you are using metrics from test data to begin with. Please explain this part a bit more clearly.

- L240: "...and thus do not include it" - should be fairly simple inclusion and I do not see why it must be excluded like this? Also, as a reader I do not know what "their construction is" - if it is so similar, please briefly explain how they do it.

---

> ### Author Response · Authors · 2024-11-18
> **Response [1 / 2]**
>
> Thank you for your detailed feedback and for taking the time to review our work. We understand that the significance of our findings may not have been fully apparent, and we appreciate the opportunity to clarify the key contributions of our paper.
>
> ---
>
> >The authors' framing of certain works as "concurrent" appears to minimize substantial overlaps, particularly with [1] and [2] which preceded the ICLR deadline by 4 and 8 months respectively.
>
> It was not our intent to minimize overlaps with these works. Our work was originally released on arxiv two weeks after [1], but this is of course a while ago now.
>
> Regarding overlap, we acknowledge that past works have pointed out issues of distribution shifts for MI evaluations. However, these works have all focused on one specific type of shift present in one or two datasets (temporal wiki and arxiv).
> We go further and present a systematic analysis of biased members and non-members across 9 published MI evaluation datasets of three different types.  We show with experiments that the shift is so severe, that any MI evaluation scores on these datasets cannot be trusted. We also show, more alarmingly, that this issue pervades other kinds of datasets - specifically our other two types of dataset constructions (biases in data replication, distinguishable tails) that are not constructed based on a temporal split.
>
> We will clarify our contributions and relationship to prior work.
>
> >The conclusion (L413) takes a problematic stance by seemingly absolving model trainers of accountability.
>
> We do not intend to suggest that model trainers should be absolved of any kind of accountability. It is not clear to us where our paper suggests this. Our point is mainly that performing MI evaluations as they are done today provides no signal.
> What we meant to say in L413 is that if researchers want to show they have a strong MI attack, they cannot use mostfoundation models to demonstrate this (unless we find a better way to construct member and non-member datasets).
>
> >The claim on L464-465 about "no knowledge of the model" requires clarification.
>
> A MIA necessarily requires the attacker or auditor to be able to interact with the target model. In contrast, our blind attack directly distinguishes members and non-members based on intrinsic differences in features.
>
> >L101: "Many of these" - Please be exact. Consider adding a table that talks about MI attack attempts on foundation models, the benchmarks they use (propose or new), and whether they suffer from the train-test split issues that the authors mention here.
>
> Please refer to Section 4 and Table 2, which provide a detailed report on the most effective MI attacks on foundation models. All the datasets listed in the table encounter issues related to train-test splits, as evidenced by the high scores achieved by the blind attack.
>
> >L154: The assumption about future dates is oversimplified and overlooks legitimate cases in fiction, climate research, and policy documents
>
> Yes, that’s why we say it is a heuristic. Our goal here is to create a blind MI attack with high TPR at low FPR. This heuristic is enough to do this (although it does have some false-positives as the reviewer suggests). It is not clear to us why this is an issue?
>
> >Some of the "blind" baselines rely on detecting data references like 2024 etc. As an adversary cognizant of the training cutoff (or even an auditor), a simple solution to fix this shift would be arbitrarily replacing all such data references for non-members
>
> The point of our blind attacks is not to obtain the highest possible performance on membership inference in a robust manner. The point of our blind attacks is to highlight that evaluation datasets that are popularly being used are not suitable for evaluating membership inference attacks.
> These edits would indeed make the specific blind baseline we consider weaker, but it would not fix all distribution shifts and so a different blind baseline would likely still work.

---

> > ### Author Response · Authors · 2024-11-18
> > **Response [2 / 2]**
> >
> > >In Table 2, why are some of the entries missing values, or entire metrics (AUC ROC) not reported? These datasets are publicly available (which is how the authors get their attack's results to begin with) so I do not see why corresponding values cannot be filled in for existing works.
> >
> > They are missing because no existing work reports that particular metric or provides a means to reproduce that metric using their attack. The column with missing entries is supposed to report “BEST Reported MIA” in the literature. It is only sensible to leave it blank if there is nothing reported in the literature at all.
> >
> > >I am confused by the 'Greedy rare word selection' protocol- is the sorting done using the TPR/FPR ratios on test data? If so, one can always design and selective use metrics to get good performance on test data if you are using metrics from test data to begin with. Please explain this part a bit more clearly.
> >
> > No, the sorting is not done on the test data and you are absolutely right that if we did that we could always easily get a good score. In each iteration of our attack, we split the dataset in a train-test split (90:10) and perform this greedy selection of rare words using only the training subset. The metrics are evaluated on the held-out test set. We repeat our experiment 10 times and report the mean.
> > This is mentioned in line 204 of our paper. We will further clarify this important point.
> >
> > >L240: "...and thus do not include it" - should be fairly simple inclusion and I do not see why it must be excluded like this? Also, as a reader I do not know what "their construction is" - if it is so similar, please briefly explain how they do it.
> >
> > We did not include this dataset because it is almost the exact same dataset as BookMIA (already discussed in our paper). This dataset starts with all the members of BookMIA and adds a few more books using the exact same concept as the construction of BookMIA—i.e., books published before a particular cut-off date are included as members, and books published after the cut-off date are considered non-members. We did test our blind baselines on this dataset and confirmed that the results were the same, which is why we chose not to include it.

---

> > > ### Comment · Reviewer_f6sv · 2024-11-18
> > >
> > > > They are missing because no existing work reports that particular metric or provides a means to reproduce that metric using their attack.
> > >
> > > Most of these attacks have open-source implementations and corresponding datasets that are available.
> > >
> > > > We did test our blind baselines on this dataset and confirmed that the results were the same, which is why we chose not to include it.
> > >
> > > Good- please mention these details and this result somewhere in the paper

---

> > > > ### Author Response · Authors · 2024-11-18
> > > > **Response**
> > > >
> > > > > The fact that it exists for multiple other models and datasets is sufficient to show how the current pipeline of privacy evaluation is flawed.
> > > >
> > > > Yes, we're glad we agree that showing this phenomenon for multiple datasets is important. This is exactly what our work does.
> > > >
> > > > > The attacks here are not truly "blind" - you are inspecting the train and test data.
> > > >
> > > > It is unclear to us how an "attack" could be doing any less than this. We are essentially learning how to distinguish members and non-members (in a generalizable way, without over fitting).
> > > > Are we supposed to show we can beat SOTA MI attacks without any knowledge of the model *or* data?
> > > >
> > > > > Yes, that was exactly my question- how much would it help? Can you still design "blind" attacks that work or is the distribution shift heavily dependent on this explicit mention of dates.
> > > >
> > > > Yes, our bag-of-words attack outperforms prior MI attacks even in the absence of explicit dates. We omitted this result as it is less interpretable but we are happy to add it if the reviewer believes it helps illustrate our point.
> > > >
> > > > > Most of these attacks have open-source implementations and corresponding datasets that are available.
> > > >
> > > > Yes, but they were clearly not intended or optimized to be used with these metrics, as the original papers don't mention them. So we don't think a comparison would be fair. If anything it would weaken our point.

---

> > > > > ### Comment · Reviewer_f6sv · 2024-11-23
> > > > >
> > > > > I think Reviewer zRbN's recent comment (https://openreview.net/forum?id=BXMoS69LLR&noteId=BFFdUJqYGt) accurately captures my feelings about this work. I do agree that there is a problem in the way membership evaluations have taken place in foundation models, and it needs to change. However, this has been explored and discussed by other papers already and while this work does add some value in exploring *how* bad it is, it is very incremental (if we already know that non-member selection is flawed, does it really matter whether it is "very flawed" or "somewhat flawed"; the conclusion either way is to look for alternatives).

---

> > ### Comment · Reviewer_f6sv · 2024-11-18
> >
> > "An alternative avenue is to forego MI evaluations on arbitrary foundation models,..." explicitly recommends not using membership inference evaluations on foundation models, without any viable suggestion *specifically* for those foundation models.
> >
> > > However, these works have all focused on one specific type of shift present in one or two datasets (temporal wiki and arxiv).
> >
> > Yes, but the final takeaway about membership inference evaluations being broken for LLMs remains the same. In that aspect, this paper's contributions are limited to showing the same result on some more datasets. The fact that it exists for multiple other models and datasets is sufficient to show how the current pipeline of privacy evaluation is flawed.
> >
> > >  In contrast, our blind attack directly distinguishes members and non-members based on intrinsic differences in features.
> >
> > This is not true, as other reviewers have also pointed out rightly. The attacks here are not truly "blind" - you are inspecting the train and test data, knowing which is which, to pick certain identifiers that are useful in distinguishing between the two.  While in theory this should be no better than random guessing, calling it "blind" is misleading.
> >
> > > These edits would indeed make the specific blind baseline we consider weaker, but it would not fix all distribution shifts and so a different blind baseline would likely still work.
> >
> > Yes, that was exactly my question- *how much* would it help? Can you still design "blind" attacks that work or is the distribution shift heavily dependent on this explicit mention of dates.

---

### Meta-Review · Area_Chair_ER7h · 2024-12-22

**Metareview:**

The authors evaluate membership inference attacks against foundation models, and find that existing attacks are ineffective for determining the membership of a given sample. In particular, the author find that a blind baseline that distinguishes between member and non-member distributions achieves higher success rate compared to existing attacks.

Reviewers generally found the message of the paper to be important and timely. However, there exist prior and concurrent work that made similar discoveries, and while the paper's message is impactful, the paper currently lacks depth and would greatly benefit from designing a practical solution. AC agrees with the reviewers and recommend rejection, but encourage the authors to improve the paper's technical depth and resubmit it to a future venue.

**Additional Comments On Reviewer Discussion:**

Reviewers and authors discussed concerns such as the claim of concurrent work, the exact definition of blindness, and practical implications of the paper's message. These concerns remain even after the author rebuttal.

---

### Decision · Program_Chairs · 2025-01-22

Reject